# Recent Advances and Synergistic Effects of Non-Precious Carbon-Based Nanomaterials as ORR Electrocatalysts: A Review

**DOI:** 10.3390/molecules28237751

**Published:** 2023-11-24

**Authors:** Laksamee Payattikul, Chen-Yu Chen, Yong-Song Chen, Mariyappan Raja Pugalenthi, Konlayutt Punyawudho

**Affiliations:** 1Department of Mechanical Engineering, Faculty of Engineering, Chiang Mai University, Chiang Mai 50200, Thailand; laksamee_p@cmu.ac.th; 2Energy Harvesting and Storage Laboratory, Mechanical Engineering, Chiang Mai University, Chiang Mai 50200, Thailand; 3Department of Mechanical Engineering, National Central University, Taoyuan 320317, Taiwan; chenyuchen@ncu.edu.tw; 4Advanced Institute of Manufacturing with High-Tech Innovations, Department of Mechanical Engineering, National Chung Cheng University, Chiayi 62102, Taiwan; imeysc@ccu.edu.tw

**Keywords:** carbon electrocatalyst, heteroatoms, single-atom metals, metal–organic framework, energy storage, oxygen reduction reaction

## Abstract

The use of platinum-free (Pt) cathode electrocatalysts for oxygen reduction reactions (ORRs) has been significantly studied over the past decade, improving slow reaction mechanisms. For many significant energy conversion and storage technologies, including fuel cells and metal–air batteries, the ORR is a crucial process. These have motivated the development of highly active and long-lasting platinum-free electrocatalysts, which cost less than proton exchange membrane fuel cells (PEMFCs). Researchers have identified a novel, non-precious carbon-based electrocatalyst material as the most effective substitute for platinum (Pt) electrocatalysts. Rich sources, outstanding electrical conductivity, adaptable molecular structures, and environmental compatibility are just a few of its benefits. Additionally, the increased surface area and the simplicity of regulating its structure can significantly improve the electrocatalyst’s reactive sites and mass transport. Other benefits include the use of heteroatoms and single or multiple metal atoms, which are capable of acting as extremely effective ORR electrocatalysts. The rapid innovations in non-precious carbon-based nanomaterials in the ORR electrocatalyst field are the main topics of this review. As a result, this review provides an overview of the basic ORR reaction and the mechanism of the active sites in non-precious carbon-based electrocatalysts. Further analysis of the development, performance, and evaluation of these systems is provided in more detail. Furthermore, the significance of doping is highlighted and discussed, which shows how researchers can enhance the properties of electrocatalysts. Finally, this review discusses the existing challenges and expectations for the development of highly efficient and inexpensive electrocatalysts that are linked to crucial technologies in this expanding field.

## 1. Introduction

It has become urgent to find new, efficient, clean energy sources due to the impact of environmental ruin and energy depletion caused by the use of fossil fuels [1]. Proton exchange membrane fuel cells (PEMFCs), which make electricity through an electrochemical process involving hydrogen (H_2_) and oxygen (air), have many benefits over conventional Li-ion batteries, including practically equivalent performance, a high energy/power density, rapid refueling, and a lack of CO_2_ emission [2]. Researchers are becoming increasingly interested in the oxygen reduction reaction (ORR), which is a crucial electrode reaction that occurs in fuel cells and rechargeable metal–air batteries to convert chemical energy into electrical power [3]. However, the efficiencies of fuel cells and metal–air batteries are severely constrained by the slow ORR reaction mechanism [3]. In the past, research has shown that there are two potential routes for the ORR process: a direct four-electron (4e^−^) pathway and a slow two-electron (2e^−^) process within standard potentials [4]. Direct four-electron activity seems to exhibit superior activity over the use of unfavorable two-electron reactions, which indicates that designing efficient electrocatalysts is required to facilitate good ORR [4,5]. In view of the search for excellent electrocatalytic performance, platinum (Pt)-based nanomaterials show outstanding potential as ORR electrocatalysts [6]. However, significant efforts have been made to discover non-precious metal (Pt-free) electrocatalysts to replace Pt-based electrocatalysts due to the disadvantages of the cost, scarcity, and deprived stability of Pt [7]. Carbon-based electrocatalysts, including heteroatom-doped carbon electrocatalysts, [8] single-metal-atom-based carbon electrocatalysts (M-N-Cs), [9], and metal–organic frameworks (MOFs), have also been explored significantly in the past decades [10] due to their inexpensive cost and outstanding electrocatalytic efficiency. Since it alters the electron density of carbon atoms within the region of heteroatom dopants, the heteroatom doping of carbon has been accepted as a successful method to adjust the characteristics of carbon nanomaterials and augment ORR performance [11]. It may also alter the chemisorption of reactants and control the chemical properties of electrocatalysts. Via the strategy of doping heteroatoms, numerous groups have recently attempted to boost the ORR activity of carbon-based nanomaterials such as carbon nanotubes and graphene [12]. The structural, electronic, and electrochemical characteristics of carbon nanomaterials have been successfully modified via heteroatom doping [13]. Graphene-doped sulfur (S), boron (B), and nitrogen (N)-doped carbon nanotubes, as well as N-doped graphene, have all been the subject of extensive research in the past [14,15,16,17]. Additionally, N- and S-doped graphene has proven to be an extraordinary non-metal ORR electrocatalyst. Furthermore, heteroatom doping has produced exciting characteristics that have led to increased attention on the imperfections of carbon nanomaterials.

According to the second law of thermodynamics, natural carbon nanomaterials, such as carbon nanotubes, graphite, and graphene, have always exhibited numerous disordered structures or imperfections on their surfaces or the edges of carbon [18]. Such defects, which are caused by the absence of specific atoms or an adjustment within the crystalline structure, definitely disrupt the electron–hole pairs and increase the ORR activity. According to the results of physical characterization and density functional theory (DFT), heteroatom doping provides edge carbon atoms with a higher electron density than pure carbon atoms, which have a similar effect.

Because of their improved ORR performance, single-metal-atom-based carbon nanomaterials (M-N-Cs) have been found to be extremely good electrocatalysts for replacing Pt-based nanomaterials [19]. Due to the importance of reducing metal particle sizes and controlling their dispersion on carbon supports, a wide range of methods have been used to prepare relatively stable and superior ORR-active metal-based electrocatalysts. Such techniques enhance the interplay between supports through the formation of chemical bonds between the metal and its related interface and the carbon support interface as well as enhance the charge carrier between both the carbon framework and metal species. The precise active centers in the metal-enclosed electrocatalyst could alter ORR activity and stability and, finally, increase ORR [20]. Metal atoms are typically attached to nitrogen sites, which noticeably changes the electronic structure of carbon atoms. Because of their exceptional electrocatalytic activity, better selectivity, high stability, and maximum atom exploitation efficiency, single-atom electrocatalysts (SAECs) have a great potential for substituting Pt-based electrocatalysts in ORR, according to DFT and experimentation calculations [19]. Three crucial elements (N, C, and M) are simultaneously present in the metal–organic frameworks (MOFs), which are chemically and structurally distinctive. Researchers now exploring them as effective precursors present the single or bi-metallic active sites, which are atomically distributed within nitrogen structures and stabilized by carbon supports [21,22,23]. The M-N-C electrocatalysts have the notable performance improvement due to these unique features. Various scientific studies of SAECs show their excellent ORR performance and comparability to the commercial Pt electrocatalyst, which have the outstanding catalytic reactions in the field of electrocatalysts. Additionally, there are significant efforts being made to create a wide range of electrochemical applications, such as fuel cells, metal–air batteries, electrochemical water splitting, solar cells, and supercapacitors [8,10,11,24].

In this review work, we start by discussing the active site mechanisms of non-precious carbon-supported or platinum-free electrocatalysts. In the second part, we briefly describe the most recent strategies for improving the ORR electrocatalytic activity of heteroatom doping, single-metal atoms (M-N-C), and MOF-based carbon electrocatalysts. Finally, we briefly go over the problems currently being experienced and the anticipated future of non-precious porous carbon-based electrocatalysts. The importance of non-precious carbon-supported electrocatalysts is highlighted and provides the pathway development of next-generation energy systems. Figure 1 shows the advantage of carbon-based electrocatalysts.

## 2. Fundamentals of Electrocatalysis Kinetics

Electrochemical applied potentials have a significant impact on ORR reactions and their kinetics. The thermodynamic potential of 1.229 V vs. standard hydrogen electrode (RHE) for ORR is indicated as a specific potential in an electrochemical reaction. The least standard potential for the chemical reaction to take place while the reactant gas is delivered is regarded as the onset potential at the anode (small) and cathode electrodes (high). Over-potential (η) is described as the additional potential present in addition to the standard potential during reactions. Figure 1a shows the ORR/OER polarization curves, which are very helpful for achieving a low over-potential with a high electrocatalysis current. Therefore, the main activity criterion in ORR reactions is half-wave potential (E half wave). For instance, the potential needed in the OER and HER reactions to achieve a current density of 10 mA cm^−2^ (Ei) is a factor used to calculate the electrocatalytic activity. The reactions of electrodes might be optimized through charge transport, and the mass transfer parameter is not considered [25]. Such currents are dependent on the applied potential, as explained by the Butler–Volmer equation. When η is greater (>0), the Butler–Volmer equation, which would be expressed in Equation (1), can be used to influence the cathode or anode reaction.
(1)$$\log(j)=log(j_{0})+\alpha zF/(2.303\;RT)\eta$$

In Equation (1), the term $\frac{\alpha zF}{2.303\;RT}\;$ is the Tafel slope. The smaller Tafel slopes contribute to the higher current density of electrocatalysis with the same *η*. The current density (*j*_0_) is calculated through the Tafel plot (*η* = 0).

## 3. General ORR Mechanism

The ORR reaction is classified into two types: the four-electron (4e^−^) and two-electron (2e^−^) pathways. During the four-electron pathway, O_2_ is converted to H_2_O (4*H*^+^ + 4*e*^−^ + *O*_2_→2*H*_2_*O* in acidic media) or OH^−^ (2*H*_2_*O* + 4*e*^−^ +*O*_2_→4*O**H*^−^ in alkaline media). In another case, O_2_ is converted to hydrogen peroxide (H_2_O_2_) following the two-electron (2e^−^) pathway that is not required for fuel cells application. Depending on the oxygen dissociation capability of the electrocatalyst surface, ORR took place via various mechanisms (dissociative or associative) in a four-electron pathway [24].


**Dissociative mechanism:**
1/2 O_2_ + ∗→ ^*^O(2)
^*^O + H^+^ + e^−^ → ^*^OH(3)
H^+^ + e^−^ + ^*^OH → H_2_O + ∗(4)



**Associative mechanism:**
O_2_ + ∗ → ^*^O_2_(5)
^*^O_2_ + H^+^ + e^−^ → ^*^OOH(6)
H^+^ + e^−^ + ^∗^OOH → ^∗^O + H_2_O(7)
^*^O + H^+^ + e^−^ → ^*^OH(8)
H^+^ + e^−^ + ^*^OH → ∗ +H_2_O(9)


The above reaction applies to acidic medium and ∗ represents active sites. In alkaline condition, the H^+^ is substituted with water (H_2_O). For direct four-electron pathways,
O_2_ + 4e^−^ + 2H_2_O → 4OH^−^(10)

It is noted that the two pathways generate the ^*^O intermediate in the ORR reaction, as shown in Figure 1b.

The mechanism referred to as the series 2-electron pathway, or alternatively known as the indirect four electron (4e^−^) pathway, involves the formation of H_2_O_2_ as an intermediary, as seen in Figure 1c. Hydrogen peroxide is first produced by a two-electron (2e^−^) route, which is followed by further reduction to H_2_O with the addition of an additional two electrons.

Distinguished researchers Griffith, Pauling, and Bridge discovered the three distinct type of O_2_ adsorption [27,28,29]. The Griffith model shows the existence of a strong bond between the atoms of an unfilled dz_2_ orbital as well as the σ orbital causing the interaction of an oxygen molecule (O_2_) with a single atom on the surface of the metal substrate [30]. The necessary electron density in the σ orbital to the dz_2_ acceptor orbital is created by end-on adsorption through the single-type bond of the oxygen molecule, according to the Pauling model. For the Pt-based electrocatalyst, the Bridge model describes how two strong bonds interact with two different active sites.

## 4. Performance Analysis of ORR Electrocatalyst

The common approaches to investigate the potential of ORR electrocatalysts is the evaluation of power density through membrane electrode assembly (MEA). However, so much work is required to construct the improved electrodes for the intensive research of ORR electrocatalysts through MEA. As a consequence, the performance of the electrochemical cell has been examined using the two-half-cell method. Both rotating ring disk electrodes (RRDEs) and rotating disk electrodes (RDEs) enable the disk electrode to rotate at a great speed of 1600 rpm, which can be used to transfer mass [31,32,33,34,35]. The screen-printed carbon electrode (GCE) surface was shielded by a fine layer of electrocatalytic materials, confirming laminar flow during rotation and lowering the resistance of mass transfer via Nafion [36,37,38]. The atomically thin and homogeneous electrocatalytic can be coated on the GCE of RDEs to investigate the ORR reaction through polarization curves for accurate measurements.

### 4.1. RDE and RRDE Method

The RDE and RRDE prototype is a useful tool for evaluating ORR electrocatalyst efficiency and also avoiding the insufficient oxygen transport [39]. Three distinct regions with a variety of reaction kinetics can be identified in the ORR polarization curve (diffusion, kinetic, and mixed kinetic diffusion controlled) of the RDE measurement. These associated ORR parameters were calculated via these equations (Equations (11) and (12)) with onset potential (E_onset_), limited current (J_L_), kinetic current density (J_K_), half-wave potential (E_1/2_), hydrogen peroxide yield (H_2_O_2−_%), electron transfer number (n), double-layer capacitance (C_dl_), electrochemical surface area (ECSA), resistance (R), and so on.
(11)$$\frac{1}{J}=\frac{1}{J_{K}}+\frac{1}{J_{L}}$$
J_L_ = 0.62 nFC_0_(D0)^2/3^ ν^−1/6^ ω^1/2^(12)

In Equation (12), ‘F’ is the faraday constant; ‘ω’ is the angular velocity (rad/s); ‘D_0_’ and ‘C_0_’ indicate the oxygen diffusion coefficient and oxygen bulk concentration, respectively. ‘ν’ is the known kinetic viscosity.

### 4.2. Membrane Electrode Assembly (MEA) Test

The fuel cell test that contains the two-electrode and electrolyte system is shown in Figure 2a,b, which reveals how the electrocatalyst is used in actual devices [40]. Electrocatalytic layers, which have been composed of electrocatalysts, carbon black, and binder, are often needed for MEA analysis. The electrocatalyst-coated membrane (ECCM) or electrocatalyst-coated substrate (ECCS) techniques are presently used to produce MEA. In general, the US Energy standard equations were used to correct the fuel cells’ current density [41,42,43]. The MEA methodology can display the real condition of fuel cells with electrocatalysts. However, its complicated fuel cell components, ambiguous electrocatalyst active sites, as well as slow mass transport in electrocatalyst layers describe the complexity associated to assessing the specific activity of electrocatalysts [44].

## 5. Active Sites Engineering

### 5.1. Active Sites of Hetero Atoms Doped Carbon Electrocatalysts

To increase the ORR activity of the electrocatalytic site and also reduce its required amount of noble metals in electrocatalysts, the common strategy of alloying methods is employed to optimize the crystal phase and structure. These processes might alter the surface adsorption, and the electronic structures in electrocatalytic sites supports the features of electrocatalysis. Pure carbon exhibits a good OOR, but nanomaterials with various metals show strong electrocatalytic activity. N, P, B, O, and S within carbon resulted in active electrocatalysts for ORRs [47]. The doping atoms generate the efficient active sites in electrocatalytic materials by altering the spin density and charge distribution of adjacent carbon-supported atoms, thereby improving its overall adsorption/desorption property. This is possible due to the carbon atoms’ distinct bond strength, electro-negativity and atomic sizes. It demonstrates that the carbon atoms in graphitic N-doped carbons that are close to nitrogen dopants contain pyridinic components that may function as electrocatalytic sites in ORR reactions [48]. Recently, both experimental and theoretical study indicate that doping creates natural carbon defect sites associated toward the material’s ORR activity, as shown in Figure 3a [49]. According to studies, the N-doping strategy was not the only factor contributing the graphene mesh with an N-dopant to electrocatalyst ORR; topological defects were also a factor [50]. Significantly, edge effects (defect sites or dopants near the corner of the electrocatalytic material) also exhibit improved activity [51]. For carbon materials with an undoped structure, a defect mechanism creates the reactive groups and is used to release the electrons from the carbon materials for the electrocatalysis process. Recently, a few carbon materials with naturally defective sites and dopant-free structures had also demonstrated great electrocatalytic activity for ORR reactions [52]. The experimental and theoretical work revealed that the tetrahedral defects that were produced in pyrolytic graphite were not entirely accountable for the electrocatalytic active sites but rather showed higher activity than N-doped pyridinic graphite [53,54]. As a result of the effect, a crucial area for future research on defects, doping, and edge sites helps to create outstanding carbon-supported electrocatalysts with distinct structural characteristics.

### 5.2. Active Sites of M-N-C Carbon Electrocatalysts

Due to atomically dispersed atoms on single-atom electrocatalysts (SAECs), usually the metal atoms contained in carbon support materials can function as effective active sites for ORR reactions. So many metal SAECs (Fe, Ir, Ru, and Mo) have been created for various electrochemical processes [54]. With the help of supporting materials like metal oxides and carbons, the created metal atoms might perform much better as electrocatalytic active sites (Figure 3b). Their improved electrocatalytic properties result in an effective interaction between both the metal and carbon support, which altered the intrinsic coordination structure of the support materials, enabling the electrons to transfer to the metal atoms and changing the adsorptive energies of the reaction products. The structural coordination and kinetics of various electrocatalytic reactions can be strongly affected by the characteristics of the support materials (nanostructure, composition, and preparation method). Likewise, the bonding of foreign atoms (S, B, and N) at preferential sites of carbon support might alter the materials’ electronic characteristics and boost the activity of electrocatalysts. According to the findings, M-N-C sites on noble metal-free carbon supports outperform Pt-based nanomaterials in ORR activity [55]. The geometrical as well as edge defects sites in the carbon support have induced the beneficial effects in a variety of single-atom electrocatalyst. The defect sites might also immobilize the metal centers from creating more defect sites and improve the support’s electrochemical and surface reaction properties [56].

### 5.3. Mechanism of Reaction at ORR Active Sites of Metal-Based Carbon Electrocatalyst

In both zinc–air batteries and fuel cells, the ORR is a crucial cathode reaction. The efficient operation of fuel cells is hampered by the numerous protons and electron transfer processes and also involves a slow kinetics reaction. Because of its outstanding reaction, platinum (Pt) is typically recognized as an effective ORR electrocatalyst. However, due to its high cost and scarcity, efforts have been made to prepare Pt-free electrocatalysts. In particular, the carbon support comprises transition metal atoms (M-N-C) with an N-dopant, where M stands for Fe, Ni, Co, etc. [57,58,59,60,61,62]. It is also possible to improve the platinum activity by adding other inexpensive transition metals atoms through altering their surfaces. Most likely, the material to substitute Pt electrocatalysts is one which is made of Fe-N-C. The active sites are represented by an M-N_x_ group as in standard nomenclature of M-N-C electrocatalysts. According to studies, the electrocatalytic activity of the M-N_x_ group in the ORR reaction increases in the following order: Fe > Co > Cu > Mn > Ni [58]. Simultaneously, interest in carbon-supported electrocatalysts has increased as a result of the discovery of N-doped carbon nanotubes in 2009, having the potential to function as extraordinary active ORR electrocatalysts [59]. The creation of electrocatalytic ORR adsorption sites is typically attributed to defects (carbon vacancies, local, and edges imperfections) in carbon support caused by heteroatoms [60]. However, recent research has not precisely pinpointed the active sites of carbon-supported materials and their instability in acidic environments. Additionally, the results obtained from the majority of studied electrocatalysts, including platinum-based electrocatalysts, reveal the necessity of high overpotential to activate the ORR. Because of the reaction intermediates, adsorption energies have a significant link. The optimization of intermediate adsorption reaction energies with another active site was not possible during an ORR [61]. Two reactions were able to establish the ORR processes at the metal site (M). The initial oxygen adsorption on the surface sites produces M-O_2_ species; afterwards, the different reactions occurred depending on the metal type or the M-O_2_ dissociation characteristics. By estimating the oxygen dissociation barrier of electrocatalysts, the ORR reaction mechanism can be classified as a dissociative or associative pathway. M-O_2_ dissociates into two M-O groups via a neighboring metal atom in the dissociative type, and each M-O group later reacts with electrons as well as protons to yield H_2_O as well as M-OH. M-O_2_ reacted with an electron and proton in an associative type to produce M-OOH. After that, M-OOH leads to H_2_O via a direct two-electron pathway (2e^−^) and to H_2_O_2_ via a four-electron pathway (4e^−^), or it creates M-OH and M-O simultaneously within the active center. Because of its outstanding performance, the last route is beneficial for fuel cells and metal–air batteries. Distinct ORR mechanisms were suggested for the various kinds of O_2_ adsorption. If oxygen connects to the electrocatalyst sites end, an associative type of ORR was used. In another case, when O_2_ is attached to the electrocatalyst sites side because the bond between the O-O group is weakened due to a π* orbital, the ORR became dissociative [62]. Researchers developed an appropriate theoretical model to analyze and examine the various parameters in the ORR reaction process (Figure 3b) [63,64,65,66,67].

**Figure 3 molecules-28-07751-f003:**
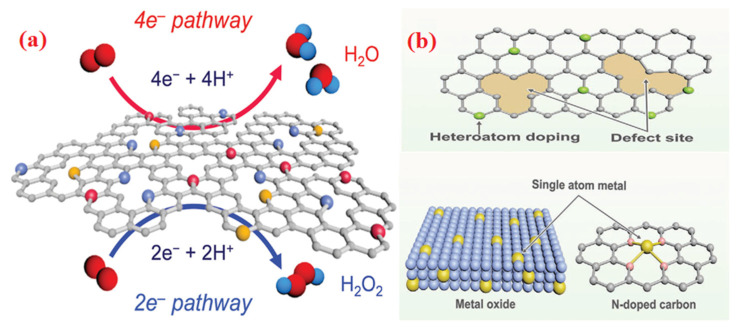
(**a**) Two pathways of ORR on heteroatom-doped carbon electrocatalysts, scheme adapted with permission from ref. [68] Copyright @ 2021, Royal society of chemistry. (**b**) Doping, defect sites and atomic metal active sites in carbon materials, scheme adapted with permission from ref. [69] Copyright @ 2020, John Wiley & Sons, Inc.

## 6. Recent Development of Non-Precious Metal Electrocatalyst

### 6.1. Metal-Free Heteroatom Dopants Electrocatalyst

Promising electrocatalysts for ORR are thought to be metal-free or carbon-supported nanomaterial; the structures are doped with nitrogen (N), phosphorus (P), boron (B), and other carbon elements [40,70,71,72,73,74,75,76,77,78]. The carbon-supported nanomaterials are generally acquired by biological systems. Advantages include a reduced cost and plentiful sources; they also have the ability to stop poisoning and corrosion, extending the lifespan of the transition metal group. The carbon structure (metal-free) and also its bonding with a heteroatom are thought to act as effective processing centers in the ORR reaction [40]. Here, we discuss the synergistic function of the heteroatom-doped carbon nanostructure for the ORR.

The charge and spin densities around at the carbon nanostructures could be altered by incorporating heteroatom dopants, which speed up the electron transport in ORRs. Numerous carbon-supported nanostructures, including graphene, carbon nanotubes, and graphene aerogel, have been thoroughly studied. An efficient ORR electrocatalyst based on carbon nanotubes was fabricated via modifying with an N-dopant. As shown in Figure 4a,b, a few of these electrocatalysts display excellent ORR activity in contrast to Pt/C in alkaline and acid media with long lifetimes [70,71]. The ability to create carbon electrocatalysis with a nitrogen dopant by carbonizing nitrogen-comprised groups [72,73,74] by heat treatment in ammonia [75,76] or nitrogen-comprised gas [77,78] would be made possible only by the good lattice similarity provided by N atoms, which has an equivalent radius to carbon atoms, as shown in Figure 4c. For instance, the pyrrole group and nitrogen were combined to create the N-carbon nanotube (N-CNT) aerogel. As shown in Figure 4d [79], XPS confirmed the N-doped carbon by detecting the presence of N-pyrrolic (400.1 eV), N-graphitic (400.9 eV), and N-pyridinic (398.2 eV) groups. These demonstrated the exceptional ORR activity (0.85 V) in alkaline medium as well as current density and half-wave potential. The entire performance, which was examined using MEA, displays excellent polarization current density at 0.5 V (199.5 mA cm^−2^) and power density (103 mW cm^−2^). The polyacrylonitrile (PAN) organic resin serves as a carbon (C) and nitrogen (N) source and is used to fabricate the highly functional, three-dimensional carbon nanostructures. In acid media, these exhibit an excellent ORR activity and E_1/2_ value (0.755 V) [80].

**Figure 4 molecules-28-07751-f004:**
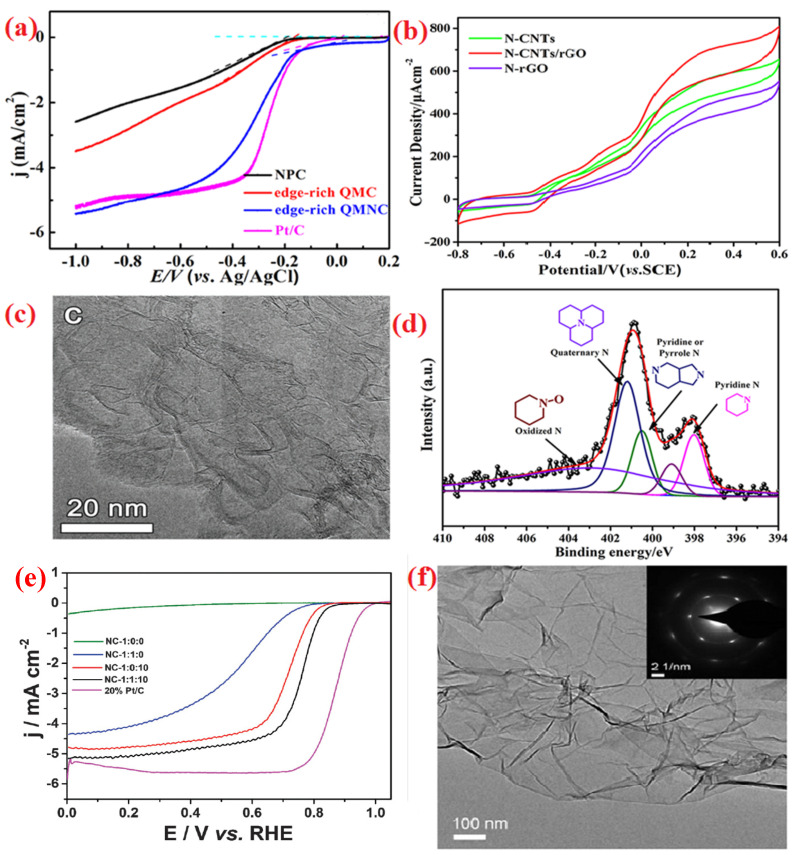
(**a**) LSV (10 mV s^−1^, 1600 rpm) on NPC, edge-rich QMC, edge-rich QMNC, and Pt/C electrodes in 0.1 M KOH saturated with oxygen, reprinted with permission from ref. [70] Copyright @ 2018, American Chemical Society, (**b**) CVs of different composites with turnover current measured in the S. putrefacient CN32 cell suspension at 1 mV s^−1^, reprinted with permission from ref. [71] @ 2018, American Chemical Society, (**c**) TEM images of LIG-O, reprinted with permission from ref. [78] Copyright @ 2018, John Wiley & Sons, Inc., (**d**) XPS spectra of N 1s spectra of N-CNT, reprinted with permission from ref. [79] @ 2015, John Wiley & Sons, Inc., (**e**) LSV curves of NC-1:1:10, NC-1:0:10, NC-1:1:0 and NC- 1:0:0 in 0.1 M HClO_4_ at 10 mVs^−1^ and 1600 r min^−1^, reprinted with permission from ref. [80] Copyright @ 2019, Royal Society of Chemistry, (**f**) TEM image of AG and SAED pattern (inset), reprinted with permission from ref. [81] Copyright @ 2013, Elsevier B.V.

Although the precise role of each carbon with N doping remains debated and needs to be meticulously studied in ORR electrocatalysts, the carbon with N doping is classified into various groups, including N-graphitic groups, N-pyridinic groups, N-pyrrolic groups, and others. To explore the contributions of nitrogen-doped groups, spectroscopy and molecule study were examined [81,82]. N-graphitic groups show superior thermal properties especially in comparison to both N-pyridinic and N-pyrrolic groups; the broad N groups are customarily easy to predict. As a result, the effect of various N doping within NGs on the ORR reaction was examined at varying temperatures. Such studies show that annealing could not change the NG morphologies and Raman spectra (Figure 4f), but it does encompass a range of oxygen contents (4.0% to 2.9%). Nevertheless, the nitrogen content gradually declines as the temperature rises. Specifically, N graphitic became increasingly prevalent while N amino and N pyrrolic were becoming less prevalent. As a consequence, the values of n and E_onset_ were determined in the following sequence: NG-1000, NG-800, NG-400, and NG-600, in which x denotes the annealed heat, which decreases with a decrease in the quantity of amino moieties after increasing with a rise in the N-graphitic content [81]. These findings show that (i) the quantity of the N-graphitic and amino group substituent determines the ORR activity (E_onset_ and n), and (ii) the amount of N-graphitic and N-pyridinic moieties determines the current density variability in the ORR, and (iii) additionally, N pyrrolic makes a negligible contribution to ORR activity. Moreover, an effective method for fabricating the carbon nanostructures with controlled nitrogen content (0 to 4.22 (at.%)) consisted of a single N-pyrrolic group (Figure 5a) [83]. To retain the N-pyrrolic group, the low-temperature carbonization treatment is crucial. Therefore, the fact that the pseudo-capacitance is entirely dependent on the N-pyrrolic content, offers a helpful suggestion for maximizing the N-dopants level and presents a novel model to investigate the ORR mechanism on nitrogen content in N-doped carbon.

The influence of nitrogen was investigated using spectroscopy analysis in addition to controlled fabrication techniques. For instance, the electronic molecule of 7,7,8,8-tetracyanoquinodimethane (TCNQ) was used to calculate the amount of transit electrons, confirming that the proportion of N pyridinic to N graphitic ([NP]:[NG]) determines the intensity of electron transfer [82] (Figure 5b). The NP: NG fraction and kinetic current density were both found to have a strong correlation. To summarize, the N-pyridinic and N-graphitic moieties may also be improving their ORR electrocatalyst performance, but their interaction may cause a decrease in the electron donation and could affect the ORR activity. The discovery provides more useful details regarding the impact of an N-doped carbon electrocatalyst on electronic and ORR activity. In addition, lighter elements such as boron, phosphorus, fluorine, and sulfur were used as carbon dopants in addition to N-doped carbon [87].

The addition of a heteroatom dopant with distinct electronegativity might rearrange the surface charge which could change the chemical and electronic properties of carbon. For instance, research was conducted on N-doped carbon with the dopant’s electronegativity (3.0) being close to or greater than that of carbon (2.5). The phosphorus (P) exhibits a lower electronegativity (2.1) than carbon, allowing it to perform a significant electron-accepting function in carbon that has been doped with phosphorus. Hou’s group created the first graphene structure doped with phosphorus using a low-cost pyrolysis method, which demonstrated superior ORR activity in alkaline media, as shown in Figure 5c [84]. The P graphene shows an excellent E_onset_ (0.92 V vs. RHE) and performance that is close to commercial electrocatalysis (20% Pt/C). In particular, graphene with P doping has better value for current density than graphene without P doping in the entire potential swiping. An investigation on developed boron (2.04)-anchored carbon could support the idea of boron-induced electron deficiency in the carbon network for an ORR reaction (Figure 5d) [85,88]. The heteroatoms of sulfur have similar electronegativity (2.58) and also experience synergistic interactions with carbon, making them a better alternative to carbon atoms [89,90]. Furthermore, this impact can significantly raise the ORR activity, particularly when adding multiple heteroatoms to carbon rather than a single heteroatom within the carbon structure. For instance, the pyrolysis-induced doping of S and N on the hierarchical carbon (NSHC) provides favorable results and motivates toward the synthesis of efficient ORR electrocatalysts. According to Figure 5e, the co-doping of S and N could significantly enhance the electrocatalytic process and afterwards induce cell performance for an ORR reaction. The co-doped NSHC structure possesses higher fuel cell performance than N-doped hierarchically carbon (NHC) in MEA conditions [86]. Additionally, MOF-5 (NPS-C-MOF-5) was used to create a ternary doping of carbon, which was then applied to ORR electrocatalysts. Dimethyl sulfoxide, triaryl phosphine and dicyandiamide were utilized as S, P and N-dopant sources during preparation. In comparison to single/binary-doped carbon electrocatalysts, derived ternary-doped NPS-C-MOF-5 has a greater positive onset potential (E_onset_) and demonstrated the excellent synergistic activity of the dopants (Figure 5f) [87]. The oxidation of graphene nanoribbons in the simple hydrothermal synthesis produced co-doped B, N graphene nanoribbons. The calculations were made to determine the doping effects of boron and nitrogen on their morphologies, configuration, size, and associated electrochemical performance. This study indicates that H_3_BO_3_ and CH_4_N_2_O in the hydrothermal process not only serve as B and N sources but also provide the active sites of electrocatalysts and enhance the synergistic effect of B and N dopant [91]. The outstanding ORR reaction is demonstrated by N-doped carbon nanosheets with a very high surface area (1793 m^2^ g^−1^) and high edge defect sites [92]. Such electrocatalysts have been found to be durable in the ORR reaction; their performance level is similar to Pt/C in an acidic environment. The DFT was employed to probe the positions and configuration of N-dopant groups, including N-pyridinic, N-pyrrolic, and N-graphitic groups, which is a crucial requirement for ORR electrocatalysts. It was noticed that the edge atoms and N-graphitic carbon atoms were significant reactive sites for the ORR reaction. The findings show that heteroatom dopants like P, O, B, and S might be combine with N-doped carbon to produce promising effects for the ORR.

The thermal treatment of polyaniline on mesoporous silica (SBA-15) resulted in N and O co-doped carbons; they were subsequently etched to remove the silica, and the researchers demonstrated that the materials have outstanding ORR reaction activity [93]. The XPS findings suggest that O and N may exist as a N-O/C-O bond within pyrrolic/pyridine groups in the carbon structure. The following conclusions come from the above discussion: for the ORR reaction, boosting the number of N sites in the carbon structure helps to generate a four-electron reaction (4e^−^) instead of a two-electron reaction (2e^−^); for the ORR reaction, increasing the electrocatalytic activity through the doping of oxygen, creating heteroatom-doped carbon structures, is very much beneficial compared to using a metal-enclosed structure. In order to achieve effective ORR activity, the researchers compared ternary co-doping this carbon structure to single or double co-doping the carbon-supported electrocatalyst (Figure 6a–e) [92]. In addition, improving the activity of the electrocatalyst in the ORR reaction and the selectivity (producing useful products like H_2_O instead of H_2_O_2_) were also vital; these were accomplished by doping the various dopants on carbon-supported materials and investigating the effects on its synergism process.

The co-doping of S and N onto carbon aerogels was obtained by using the pyrolysis of carrageenan and urea [94]. The created carbon has a lot of interconnected mesopores and macrospores as well as a lot of surface area (up to 1307 m^2^ g^−1^), which helps mass transport and forms the groups that are more reactive to the material. As a result, such materials demonstrated exceptional ORR activity in acidic media; the performance was comparable to that of commercial Pt/C electrocatalysts. Theoretical analysis suggests the N-S-C defect sites within the carbon nanostructures might act as ORR active sites. High efficient electrocatalysts were developed in N-S-D-G-4 and N-S-D-G-6 carbon structures and we obtained the reduced ORR overpotentials, which were calculated in theoretical and experiential analysis. Thus, the researchers demonstrated the significance of polyhedral S defects and graphitic-type N dopant sites within materials, as they contributed to the ORR electrocatalytic activity. These investigations further demonstrate that the improved ORR activity of the carbon electrocatalysts was influenced by the dopant sites. The creation of more active species in a carbon structure was made possible by heteroatoms, porous structures, electronic structures, and defect sites. However, the specific roles played by active sites, including defect sites, pore edges, and dopants in electrocatalysts, are still unidentified [95]. To acquire a solid understanding of how different types of active sites contributed to ORR activity, sophisticated characterization methods with a combination of meticulous preparation and DFT analysis are necessary. Additionally, these electrocatalysts may be contaminated as a result of the synthesis vessel and precursor utilized; these must be potentially avoided to reduce the effect of contaminants on the electrocatalysts’ final ORR activity. The various properties of heteroatom-doped carbon electrocatalysts are presented in Table 1.

### 6.2. M-N-C Carbon Electrocatalyst

The nitrogen is doped with carbon and then combined to form a single-atom or non-precious metal nano-structure, resulting in an M-N-C structure with highly active sites used to evaluate the voltage in the ORR both in alkaline and acidic media [100,101,102]. Numerous non-precious metal atoms were studied in the M-N-C structure such as cobalt (Co), irons (Fe), manganese (Mn), copper (Cu), and nickel (Ni), zinc (Zn), and other elements (metal atoms indicated as ‘M’). Additionally, the coordination number is altered by methods of synthesis, whilst the N and C components represent the bonding of metal atoms to nitrogen or carbon atoms. An in-depth investigation of the electrocatalysis based on the M-N-C structure revealed that the exhibited ORR reaction has high activity, selectivity, and longer life. Graphene substrates and Fe_x_N nanostructures have been developed. The resulting electrocatalyst shows superior E_onset_, greater current density, and lower charge carrier resistance (R_ct_) in alkaline solution compared to that result obtained without Fe-N-C (Figure 7a,b) [103]. A Fe-N-C catalyst (P-FeMOF@ ZIF-8) that has been made using the pyrolysis method also exhibits excellent ORR activity in both alkaline and acidic media. The atomic-level analysis of Mossbauer spectroscopy and EXAFS (Figure 7c,d) [104] verified the occupation of Fe within Fe-N_4_ coordination.

The core issue involves understanding the active sites of M-N-C single-atom structures because it is useful to gain knowledge about the reaction mechanisms and further encourage the ORR activity of electrocatalysts. Four elements, including transition metal centers with highly dispersed states, adjacent coordinated N atoms connected with carbon skeletons, heteroatom dopants as environmental atoms, and guest groups, are recognized as the M-N-C active sites in single-atom electrocatalysts (SAECs) [105]. The combination of a carbon-supported structure with transition metal nanoparticles is always considered as SAECs and nanoparticles (Table 2).

**Figure 7 molecules-28-07751-f007:**
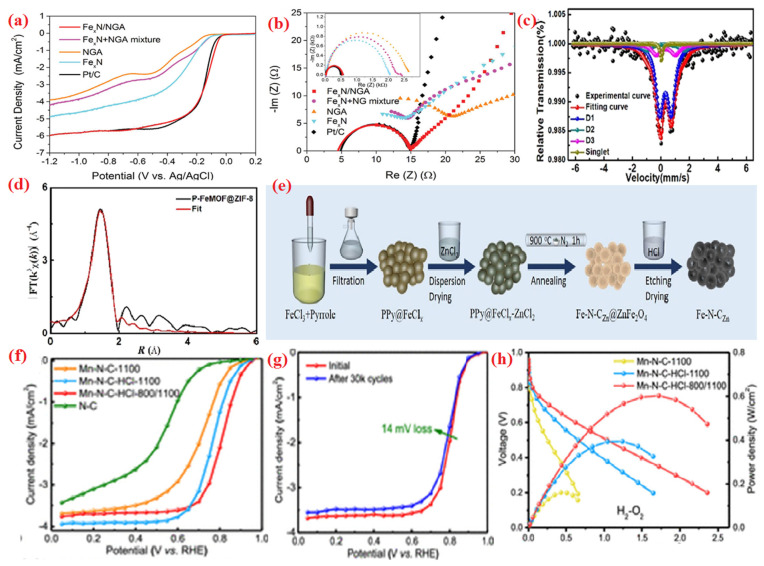
(**a**,**b**) RDE voltammograms and impedance spectra of Fe_x_ N/NGA, Fe_x_ N+NGA, NGA, free Fe_x_ N and Pt/C in 0.1 M KOH at 10 mV s^−1^ at 1600 rpm, reprinted with permission from ref. [103] Copyright @ 2014, John Wiley & Sons, Inc., (**c**) Fe Moössbauer spectrum and (**d**) FT-EXAFS fitting curve of P-FeMoF@ZIF-8, reprinted with permission from ref. [104] Copyright @ 2019, American Chemical Society, (**e**) synthesis of the micro-mesoporous Fe-N-C-Zn, scheme adapted with permission from ref. [106] Copyright @ 2019, American Chemical Society, (**f**) polarization curves of Mn−N−C-1100, Mn−N−C−HCl-1100, Mn−N−C−HCl-800/1100, and N−C, (**g**) steady-state ORR polarization curves of Mn−N−C−HCl-800/1100 before and after potential cycling tests (0.6−1.0 V, 30,000 cycles), (**h**) cell performance of different catalysts in H_2_-O_2_, reprinted with permission from ref. [107] Copyright @ 2020, American Chemical Society.

**Table 2 molecules-28-07751-t002:** Various parameters of single-atom-based ORR electrocatalyst.

Metal Precursor	Catalysts Name	Metal Content	BET Surface (m^2^ g^−1^)	Electrolyte	E_onset_ (V)	E_1/2_ (V)	Tafel Slopes (mV dec^−1^)	Electron Transfer Number	H_2_O_2_ Yield	Reference
Fe	Fe1/N,S-PC (Fe/N and S-co-doped hierarchical porous carbon)	2.6 wt.% Fe (ICP-OES)	998.5	0.1 M KOH	-	0.904	84.5	3.95	<9%	[101]
Fe	Fe−N−C (P-FeMOF@ZIF-8)	1.24 wt.% Fe (ICP)	785.0	0.1 M KOH	1.01	-	-	-	-	[104]
				0.5 M H_2_SO_4_	0.85	-	-	-	-	
Mn	Mn-N-C-HCl-800/1100	2.00 wt.% Mn (XPS)	1511	0.5 M H_2_SO_4_	-	0.815	-	-	<3%	[107]
Fe	Fe/N-PCNs	3.89 wt.% Fe (XPS)	864	0.1 M KOH	0.96	0.86	-	~3.95	-	[108]
				0.1 M HClO_4_	0.88	0.79	-	3.80		
Mo	Mo/OSG-H	13.47 wt.% Mo (ICP-OES)	-	0.1 M KOH	0.78	-	54.7	2.1	-	[109]
Fe/Co	CAN-Pc(Fe/Co)	10.70 wt.% (ICP-OES)	84.11	0.1 M KOH	1.04	0.84	54	3.94	<5%	[110]
Zn/Co	Zn/CoN-C	0.33 wt.% Zn 0.14 wt.% Co (ICP-MS)	1343	0.1 M KOH	1.004	0.861	-	3.88	~5%	[111]
				0.1 M HClO_4_	0.97	0.796	-			
Zn	Zn–N–C	9.33 wt% Zn (EDX) 5.64 wt.% Zn (ICP-MS)	1002	0.1 M KOH	-	0.873	-	-	<5%	[112]
				0.1 M HClO_4_	-	0.746	-			
Co	10Co-N@DCNF	0.649 wt.% Co (ICP-MS)	-	0.1 M KOH	-	0.83	56	-	<8%	[113]
Co	Co-N/S-DSHCN-3.5 (Co, N, and S co-doped hollow carbon nanocages)	-	429	0.1 M KOH	0.989	0.878	-	3.9	<10%	[114]
				0.5 M H_2_SO_4_	0.84	0.754	-			
Fe	FeCl1N4/CNS	1.50 wt.% Fe (ICP-OES)	-	0.1 M KOH	-	0.921	51	3.97–3.99	<1%	[115]
Fe/Co	M/FeCo-SAs-N-C	5.12 wt.% Fe 4.39 wt.% Co (ICP-MS) 3.32 wt.% Fe 3.33 wt.% Co (XPS)	1003.7	0.1 M HClO_4_	0.981	0.851	-	~4	<6%	[116]
Mn	20Mn-NC-second	3.03 wt.% Mn (ICP-MS)	715	0.5 M H_2_SO_4_	-	0.8	80	-	<2%	[117]
Co	Co-N-C-10	-	-	0.1 M HClO_4_	0.92	0.79	55.8	-	<2%	[118]
Cr	Cr/N/C-950	1.90 wt.% Cr (ICP-MS)	884.9	0.1 M HClO_4_	-	0.761	37	-	-	[119]
Fe	C-rGO-ZIF-2	4.29 wt.% Fe (ICP-OES)	650	0.1 M HClO_4_	0.89	0.77	-	3.8	<5%	[120]
Ce/Fe	Ce–Fe/NC	0.83 wt.% Ce 0.88 wt.% Fe (ICP-OES)	831	0.1 M KOH	-	0.913	55.55	3.98	~3%	[121]
				0.1 M HClO_4_	-	0.791				
Mn	Mn-N-C	-	1334.3	0.1 M KOH	0.98	0.88	60.3	3.98	<2%	[122]
				0.5 M H_2_SO_4_	-	0.73				
Mn/Co	MnCo_2_O_4_/N-C	-	630.5	0.1 M KOH	0.943	0.795	86	3.50–3.83	<10%	[123]

The transition metal site center is considered to be the key factor in determining the active sites, which have a significant impact on the SAECs’ intrinsic ORR activity. The interactions of metals (d-orbitals) with oxygen reactants species (p-electrons) modify their intermediates during the ORR process. These will aid the sorption process of oxygen molecules and facilitate the electron transport reaction [124]. Generally, transition metal groups with decent ORRs were created using cobalt (Co), iron (Fe), nickel (Ni), manganese (Mn), and zinc (Zn) [108,109,110,111,112,125]. For instance, Fe-N-C-derived carbon electrocatalysis has been created by the in situ polymerizing of iron, as shown in Figure 7e [106]. A high current density, excellent stability, and large positive E_1/2_ and E_onset_ values are accomplished in alkaline media by the Fe-N-C electrocatalyst. An atomically confined electrocatalyst with a Mn-N-C structure with closely packed Mn-N_4_ active sites was designed; this is depicted in Figure 7f,g [107]. M-N-C generally contains an Fe metal center. Figure 7h shows that the produced Mn-N-C- electrocatalysts perform much better than Pt-free ORR electrocatalysts in terms of their significant ORR activity and elevated power density (0.39 W cm^−2^) in 1.0 bar of H_2_-O_2_ gas in MEA. Discovering the ideal metal is of crucial importance and thus has been extensively examined. It was discovered that the electrocatalyst of various transition metals (Fe, Mn, Co, Ni, and Cu) had a serious influence on its structure as well as affected the ORR activity [58]. The ORR activities of different metal atoms improved in the following order: Fe > Co > Cu > Mn > Ni. The best choice of central metal in the M-N-C group can only be determined via experimental studies, since it is challenging to regulate different parameters, such as the degree of graphitic structures, morphologies, etc. Theoretical studies are an excellent option to assess the ORR activity of metal sites in M-N-C in order to overcome this challenge. To determine the fundamental activity of the M-N-C group with numerous central metals (M), the density functional theory (DFT) model seems to be an essential tool [126]. The crucial indicator of E_onset_ activity was calculated and matched with *OH adsorption energy to display a volcano slope (Figure 8a). The iron element, which is located just on slope of a volcano, exhibits an excellent ORR for M-N-C.

The coordination sites atom (N) is directly combined with the carbon structure (C) and the central metal atoms (M), which connect the C and M sites. In the M-N-C active sites, the nitrogen atom (N) is typically bonded to the central metal of four atoms via d-p and π-bonds [105]. Because of the improved performance of the ORR reaction, the nitrogen atom that is present at the M-N-C active sites was investigated. DFT is employed to compare the ORR activity of N-graphitic and graphitic carbons (Fe_3_C/NG and Fe_3_C/G, respectively) (Figure 8b) [127]. Fe_3_C/NG demonstrates the superb stability of the *O intermediate and reveals the critical function of the nitrogen element. Especially, the NG layer offers adequate support for Fe_3_C to display excellent ORR activity when compared to the Pt (111) surface, as verified by DFT study. Additionally, it is thought that the specific condition of the coordination atoms modifies their electronic structures and will affect the active M-N-C sites. By evaluating the oxygen adsorption ability using the Gibbs free energy (G) on the cobalt-anchored nitrogen-doped carbon nanosheets (Co/N-CNSNs), it was feasible to study the coordination atoms in specific chemical environments (Figure 8c). In contrast to the N-graphitic and N-pyrrolic sites, it has been demonstrated that the value of G in carbon sites close to pyridinic nitrogen exhibits the lowest G value. These demonstrate that N-pyridinic sites can absorb oxygen under favorable environmental conditions [128]. Another effective method for changing the electronic structure and M-N-C activity is to control the hierarchy of coordination atoms. These studies inspired the researcher to prepared the carbon structures with various types of defects, such as CoN_4−x_C_x_ (x = 0–4). In order to comprehend the relationship between the activity and the active sites’ structure, which corresponds to the CoN_4–x_ C_x_ structure, DFT studies were used. The results of the acquired activity are as follows: CoN_4_, CoN_2_C_2−3_, CoN_3_C_1_, CoC_4_, CoN_2_C_2-2_, CoN_2_C_2−1_, and CoN_1_C_3_ exhibit higher cyclic stability, mass activity, and excellent power density, as seen in Figure 8d–g. The improvement of the CoC_4_ configuration in all seven investigation sites is a key finding, indicating that Co and C_4_ had a greater impact than the other four coordinations that were only partially N doped (CoN_2_C_2−2_, CoN_2_C_2−1_, and CoN_1_C_3_) [113]. Additionally, the coordination of the central metal and dopant atoms had a crucial impact on the ORR properties, such as the two electron reactions and n, without experimental effects [129,130]. For instance, the two-electron process (2e^−^) was facilitated by the synthesized SAECs with high Mo loading (10 wt%), which contain S and O atoms and exhibit excellent hydroxide selectivity (95%) in alkaline solution, making them promising electrocatalysts for the ORR process (Figure 9a). As results, four-electron processes were facilitated by the particular coordination structure of MnN_4_ groups, and it is clear that the specific active sites in the M-N-C structure play a crucial role in determining the reactions pathways and n.

The atoms in the carbon structure that are not attached to M species (dopant atoms) and the N and S atoms that are linked to the central metal atoms (guest atoms) could impact the activity of M-N-C sites through the process of accepting and donating the electrons. Numerous heteroatom atoms (N, S, B, etc.) as dopant atoms were anchored within the carbon structure to alter the ORR activity [114,131,133]. The sulfur dopant was added to the Fe-N-C nanosheet in a unique way through the pyrolysis process (Figure 9b), and as a result, the Fe-N-C nanosheet exhibits superior ORR activity in the RDE method, which is thought to be due to the sulfur dopant inducing the active sites of Fe-N-C (Figure 9c) [131].

Some researchers looked at how guest atoms in the active sites affected ORR activity. By using a chlorine experimental setup, the influence of Fe-N-C active sites by chlorine atoms in alkaline media was investigated. This finding suggests that the short-range interaction of chlorine with an iron center might modify the electronic structure of active sites. As a result, as shown in Figure 9d,e [115], the ORR activity and cyclic stability of electrocatalysts in alkaline media were sufficiently improved. The active sites were stimulated, and the ORR activity was increased only when transition metal nanoparticles were combined with M-N-C. In particular, the anchored nanoparticles within M-N-C groups strongly motivate the interaction with carbon shell or sheet structures, which stimulates a synergistic interaction between M-N-C and nanoparticles that improved the ORR activity. For instance, it was suggested to investigate this effect by anchoring the cobalt within Co/N-CNSNs via DFT and a poisoning method [128]. The ORR process is aided by the conductive nature of the anchored metal nanoparticles, which are revealed by the higher ΔG of M-N-C structures without metal content. The high electronegativity of carbon shells is caused by the Co component, which helps electrons flow on the outer layer of nitrogen-doped carbon materials. In a study conducted on M-N-C nanospheres decorated with Fe/Fe_3_C in which researchers examined their electrocatalytic activity, the findings revealed the increase in the ORR activity. Such results demonstrated that the N-doped carbon and M-N-C might further modify the reactive groups in the carbon outer layer through the anchoring of iron atoms. Specifically, the connected iron atoms change the M-N-C active sites, especially on external carbon shells, which improves oxygen adsorption capacity and boosts ORR activity [134]. Further to that, it is well known that the influence of corrosion in acidic conditions leads to poor cyclic durability and inability. These cause make it difficult to identify the reactive groups in acidic media and create serious difficulties for metal nanoparticles. By creating the carbon-coated nanoparticles as depicted in Figure 9f, it is possible to increase the acidic stability of M-N-C sites. A novel strategy was successfully often used to develop a Fe-Co-N-C electrocatalyst (M/Fe-Co-SAs-N-C), wherein the carbon layer perfectly encompasses the metal atoms (Fe, Co) [116]. The synergism among nanoparticles (M) and Fe-N_4_ sites modulates the oxygen adsorbs capacity and extended O-O bond, which could aid accelerating ORR in an acid medium, which was revealed by a DFT study. The O_2_ dissociation barrier of M/Fe-Co-SAs-N-C electrocatalysts has been lowered during ORR, which may be advantageous for a four-electron reaction.

Exploring and understanding the electrocatalyst efficiency is crucial, especially for fuel cell technology and metal–air batteries because it entirely opens up the possibility of substituting with a commercial Pt electrocatalyst. The Fe-N-C groups are thought to be the most intensely investigated and highly active electrocatalysts utilized in fuel cell MEA. The isolated Fe-N_4_ active site within a carbon network structure was created and employed as an efficient electrocatalyst to serve as the PEMFC cathode [132]. These demonstrate the excellent current density in PEMFC operations and the remarkable ORR activity of E_onset_, which was about 0.906 V through acid media (Figure 9g–i). However, the existence of Fe content induces the Fenton reaction, leading to the creation of free radical groups such as hydroperoxyl and hydroxyl. These radicals deteriorated the membranes’ structure, and the poor stability was witnessed in PEMFC by Fe-N-C electrocatalysts. Mn-N-C and Co-N-C electrocatalysts are encouraging options for lowering the Fenton reactivity with satisfactory activity. The Mn-N-C electrocatalyst has effective active sites and demonstrates the better ORR activity of E_1/2_ (about 0.80 V in acid solution, which is significant compared to the Fe-N-C group). The superior corrosion resistance of the Mn-N-C electrocatalysts obtained from the Mn metal created the effective carbon structure with strong active sites offer satisfactory efficiency and long-term durability (Figure 10a–j) [117].

### 6.3. Metal–Organic Framework (MOF) Carbon Electrocatalyst

It has been widely accepted that electrocatalysts made from metal–organic frameworks (MOFs) are more effective because they have large absorbent sites, uniform pore volume, controllable morphology, good electrochemical properties as well as a topographical scaffold. Brilliant competence is still needed for energy storage and transformation because it must satisfy specific requirements for industrialization. The various MOFs and their collection have been established as effective electrocatalysts for ORR reactions. Additionally, MOF-derived electrocatalysis offers beneficial methods for boosting ORR performance as well as provides the unique fabrication techniques that enable the large number of ORR active sites. The progress of MOF-derived electrocatalysts for ORR reactions has been recently analyzed, which covers the vital properties related to the mechanism of adsorption sites including the host sites, metal sites, dopant site and their synergistic interactions [135].

The porous polymer of MOFs is created by combining the metal clusters and organic precursors produced through the porous electrocatalysts. It is interesting to note that the MOF has a variety of qualities, including controllable porosity, adaptable morphology, and easy regulation of nanostructures, all of which are promising for electrocatalysis applications (Figure 11a) [136]. The most recent ORR development addresses the connection between structural property and interactions. Trying to combine the composition of inorganic heterogeneous and molecular homogeneous electrocatalysts, MOFs may contribute to achieving high performance and can be investigated as excellent electrocatalysts [137]. The hydrothermally produced MOFs had a well-crystalline nature, which act as effective bifunctional electrocatalysts. The resulting structure has a high surface area and a rich micro-porous composition, and it exhibits two electron reactions (2e^−^) at applied voltages between −0.30 and −0.50 V. It also reveals four electron reactions (4e^−^) at voltages between −0.50 and −0.95 V, respectively [138]. A metal electrocatalyst based on MOF has been synthesized that is made up of Pt free and exhibits the improved ORR efficiency in alkaline media. Compared to Pt-C, these electrocatalysts exhibit significant ORR efficiency at half-wave potentials (0.7 V) and onset potentials (0.85 V) as well as better stability and tolerance to methanol [139] (Figure 11b–d). By the pyrolysis of the zeolitic imidazolate framework (ZIF-9) with sulfur, the porous carbon structure was created. These notable mesoporous composites with porous carbon structures that enclose nanoparticles might enhance the ORR electrocatalytic performance and present the 4e^−^ ORR pathway in alkaline media [140]. In recent years, the growth of electrocatalysts derived from MOF for energy storage as well as conversion technology has been evaluated and utilized in various applications including PEMFC, batteries, water splitting, supercapacitors, and ORRs. For extremely stable electrochemical technology application, the MOF-derived electrocatalyst with a unique structural design was carefully reviewed [141]. The electrocatalyst formed from the MOFs was found to have an effective porous structural morphology, which makes them a possible candidate for electrode applications with excellent performance. These materials provide a unique perspective to designing and creating novel electrocatalyst morphologies [142]. The fabrication and utilization of MOF-derived materials for energy storage and conversion, fuel storage, as well as hydrogen storage are covered in this study. It was explored how these materials might be used in energy applications. In comparison to commercial Pt-C, these MOFs have excellent methanol crossover barriers and long-lasting stability. This research laid the groundwork for fabricating new and relatively affordable electrocatalysts that function better in ORR [143]. There have been reports of carbon-derived materials with adjustable porous structures as well as a multidimensional core–shell structure. By subsequently surface polymerizing the identical monomers within carbon black XC-72 and silica nanoparticles, highly porous core–shell nanostructures were created. Carbon nanoparticles are like a substrate, and a spiro is like a twin monomer that might be employed to create a mesoporous carbon material core with a microporous shell structure. Moreover, applying the polymerizing effect of Spiro within silica nanoparticles led to a carbon material with hollow nanospheres featuring a double shell structure and a hierarchical arrangement [144]. A highly innovative electrocatalyst for the ORR reaction has been obtained, which was created by pyrolyzing a Co-derived polymeric organic framework. This remarkable framework demonstrated the good ORR electrocatalytic activity in acidic solution through the 4e^−^ route. Furthermore, this result is comparable to the available Pt-C without a methanol crossover. The study indicates the unique synthesis methods for creating new metal-free electrocatalysts with enhanced electrocatalytic activity for ORR in fuel cells as well as provides promising insight into ORR reactions that occurred at the M/N/C electrocatalyst. MOFs have vast surfaces because of their inner pores, but crystalline techniques could significantly identify their precise structures. However, the biggest problem with this electrocatalysis is really their lower chemical stability compared to MOFs because these reactions are carried out in electrolyte media, which typically required the highly acidic or basic condition [145]. As a result, research on MOF-based material has lagged far behind that of MOF-derived materials, including nanocomposites made of carbon, metal, and metal oxide [146,147,148,149]. Nevertheless, some MOFs exhibit improved stability in water as well as in extremely alkaline and acidic conditions [150,151]. Enhancing the interfacial adhesion between the metal and the ligand structure is crucial for designing efficient MOFs with improved electrochemical stability. Strong bond strength can be created only through the joining of metal ions with higher oxidizing states and carboxylate ligands in addition to joining soft metal ions of incredibly simple isolate ligands [152]. Dual-shelled nano-cages (DSNCs) contain an outer wall made of Co and N co-doped carbon (Co-NC) and an inner shell made of N-doped carbons, which were fabricated easily by a novel approach. The dual-shelled NC/Co-NC nano-cage is effectively constructed with Co-NC, performing more effectively and exhibiting excellent ORR properties, which are superior to those of Pt and RuO_2_ ORR electrocatalysts in alkaline conditions. The improved performance of Co-NGC is attributed to the synergistic electron transport between the Co nanoparticles as well as the N-doped carbon, as revealed by first principles analyses (Figure 11f–h) [153].

The fabrication of electrocatalysts (Fe/N_x_/C and N/Fe/Fe_3_C–C–RGO) to activate the ORR process by using the Co-based MOF, transition metal-based MOF, and Fe-based MOF were investigated [154,155,156]. Recently, one researcher investigated the shaft-like Fe-based MOFs with controllable dimensions and shapes, which were effectively made from components of Fe_3_O (H_2_N-BDC)_3_ as well as 2-aminoterephtalic acid. The porous structure of MIL-B-NH_3_ is unique in that it contains effectively distributed Fe and N atoms. The produced Fe-N_x_-C electrocatalyst showed enhanced ORR performance in comparison to Pt-C [157]. The electrocatalyst of the Co-Fe structure encased in N-graphene exhibits improved electrochemical characteristics, which was created through carbonizing a Prussian blue and cobaltocene/cobalt nitrate [158] (Figure 12a,b).

Because of their outstanding electron permeability, porous structure, and variety of application areas, carbon materials made from MOFs had recently attracted a lot of interest from the scientific community. Researchers have attempted to modify the morphological and structure characteristics of the carbon materials obtained from MOFs in order to improve their own physical and chemical characteristics. A discussion of various kinds of carbon materials obtained from MOFs with an emphasis on energy applications has already taken place [161].

An efficient electrocatalyst for ORR has been satisfactorily developed and produced, which consisted of a one-dimensional (1D) carbon rod made of S and N gathered with Ni-based MOFs [159] (Figure 12c–g). The fabrications of Ni_2_P nanoparticles anchored carbon polyhedrons (Ni_2_P-CoN/PCP) were obtained from MOFs. As a result, the Ni/P-CoN-PCP electrocatalyst exhibits exceptional functional electrocatalytic activity and promising cost-effective ORR electrocatalysts for fuel cells. This was attributed to a synergy between Ni_2_P particles and the strong active centers of Co-N_x_ in Co-N-PCP electrocatalysts, which contain high electrical conductivity with a more disordered structure [160] (Figure 12h–j). In recent times, it has been noted that N-doped graphitic carbon materials are effective ORR electrocatalysts when operated in alkaline solution. Better ORR durability and efficacy have been found in carbon-based materials by utilizing carbon nanotubes and N-doped carbon nanotubes [162].

According to earlier research, the incorporation of extremely lightly coated carbon MOFs onto GO nanosheets could even produce a layer-by-layer structure with a huge surface area and exceptional electrical properties [162,163,164]. The ZIF/8/GO material was pyrolyzed at 800 °C in attempt to develop an emerging material (Figure 13a–c). Later, 2D carbon–graphene–carbon materials were developed, which significantly shows the better ORR results. A thin-layered N-doped carbon nanostructure maintains the necessary active sites and sustained the suitable stability between subsequent electrical conductivity, which is required for ORR. Therefore, the ORR output of nanocarbon, graphene, and nanocarbon is outstanding. An increased onset potential of 0.92 V as well as a high limiting current density of 5.2 mAcm^2^ at 0.6 V was obtained in ORRs [160]. Many kinds of MOF precursors, including Zr-MOF, Cu-MOF, Cd-MOF, and polyoxometalate-MOFs, were extensively studied to generate unique metal–carbon electrocatalysts for fuel cells [165,166,167,168,169]. MoO_3_ is a rare transition metal oxide with exceptional chemical stability and electrical conductivity. It will attract more interest in this field of study. The main reason for promoting the MoO_2_ as an ORR electrocatalyst in fuel cells was due to the efficient catalytically active sites, which generally provide a the molybdenum edge and the oxygen edge in the metal oxides [170]. Researchers also studied the MoO_2_-GO materials obtained from the polyoxometalate precursors [171]. The ORR reaction in the fuel cell is more favorable only when the carbon materials are doped with cobalt or iron and then doped with nitrogen. Pt-C based electrocatalysts are replaced by these novel doped electrocatalysts. Due to the synergistic interplay among two active metal sites, it is possible to even further increase the ORR activity as compared to single-metal doping (Figure 13d,e) [172]. Electrocatalysts based on MOFs have also made major advancements in the past few years. When compared with traditional electrocatalysts, MOFs usually display less electric conductivity. In order to take full advantage of MOFs, uncover the reaction mechanisms, improve the materials’ ORR, and understand the depth of their electrochemical behavior, these electrocatalysts could be examined more. The development of MOFs-based electrocatalysts for ORR is still in its early stages, and many difficulties still need to be resolved before the fabrication of proficient MOFs-based electrocatalysts for ORRs will become a reality. Some of the studied MOFs are only available for definite applications. For instance, highly conductive and much more stable MOFs are preferred as pure electrocatalysts, and MOFs-based frameworks for metal-free materials seem to be effective electrocatalysts. The advancement from basic research to industrial growth and commercial application was severely constrained by the difficult preparation methods for some MOFs as well as the limited number of manufacturing methods. The synergistic effects stimulated the improved performance of electrocatalytic activity in the MOF composites. Based on the obtained reports, there are some pure MOFs that can be utilized as electrocatalysts. During the ORR process, the surface of the MOFs experiences an irreversible phase change. Therefore, it is necessary to identify these actual active sites using more inventive methods and the accurate controlling of active sites, particularly in MOF materials. Further theoretical research and experimental investigation are required to address the lack of knowledge regarding the process for converting MOFs into their different products with related mechanism.

Derivatives of zeolitic–imidazolate–framework-8 (ZIF-8) have been demonstrated as ideal precursors for making metal-based SAECs for ORRs. Scientists have just described a new ZIF-8 thermal treatment method to produce graphene nanosheets co-doped with Fe and N (Fe-N-GNs). This shows the half-wave potential of 0.903 V in a basic condition and 0.837 V in an acidic condition, in which the Fe-N-GNs SACs exhibit exceptional ORR activity similar to traditional Pt-C electrocatalysts (Figure 13f–h) [174]. Researchers have developed a silica-based MOF template process for the fabrication of an elongated nanostructure decorated with single-atom Fe active sites for ORRs, which shows the outstanding efficiency in both basic and acidic solution (Figure 14a–g). These extraordinary electrocatalytic results are attributed to the presence of edge-rich nanostructures containing the three-phase boundary; they facilitate the reactants to connect the Fe-active sites of neighboring single atoms. These scenarios demonstrate the feasibility of the preparation method [175]. The various types of electrocatalyst were studied for ORRs in proton exchange membrane fuel cells (PEMFCs) and energy conversion applications (Table 3).

## 7. Conclusions and Future Outlook

Technologically and scientifically, the development of high-performance Pt-free electrocatalysts is required to speed up the vast exploitation of feasible renewable power. It has been a main concern for many years to find a strong replacement for Pt electrocatalysts in fuel cells. Ingenious plans and designs have been developed to enhance ORR activity. The new carbon-based nanomaterials as ORR electrode materials in fuel cells have been revealed as excellent electrocatalysts. Electrocatalysts can be formed from carbon doping, metal-based carbon, and MOF nanostructures. Non-noble metal-based SAECs are interesting due their improved active sites, but more precise methods are needed to produce high-quality electrocatalysts. Currently, it is necessary to obtain ORR electrocatalysts with increased performance in an acidic medium compared to an alkaline medium. Thereby, the higher surface area and augmented active sites of materials are desirable, since they might lead to an improvement in the electrocatalyst’s electrochemical properties. Despite the numerous difficulties throughout this area, the growing interest in electrochemical reduction by non-noble metal electrocatalysts indicates a positive future development. Additional investigations into the reduction mechanism as well as fabrications of effective electrodes are a crucial step to take in the future. The non-noble metal electrocatalysts may therefore find extensive exploitation in fuel cell systems and metal–air batteries for effective ORRs. One of the most important objectives is the improvement of the electrocatalyst for PEMFCs. The essential purpose of electrocatalysts is to maintain the stable continuous operation in fuel cells in order to make them viable for industrial fabrication. Nevertheless, safeguarding the electrocatalyst is indeed very challenging. The development of high-performance MOF-based SAECs will undoubtedly be delayed by the complex and time-consuming fabrication techniques. Therefore, straightforward, affordable design and manufacturing methods are required to achieve the electrocatalyst’s massive scale and also satisfy the electrochemical test conditions. These will ultimately lead toward their practical applications in renewable technology [179,180]. It is necessary to overcome the current technical obstacles to fabricate useful nanostructures, which will require practical utility [181]. The theoretical calculation and simulation models reveal the hidden fact of ORR reaction mechanisms and inherent active sites, which will be evaluated further in on-going research. The experiment as well as measurement techniques have a significant impact on the creation of better ORR electrocatalysts [182,183]. In order to develop as well as prepare the ORR electrocatalysts for PEMFCs, detailed and consistent theoretical research must be implemented on nanomaterials.

### Future Efficient Carbon-Based Electrocatalyst

**Heteroatom-doped carbon electrocatalysts** contain the advantage of stability functioning in acidic environments due to the controlling of metal leaks and metal contamination. Using a co-doping model can generate active sites and quickly transfer charges. Although many metal-free electrocatalysts have been created over the years using a variety of methods, there is still much to learn about the widespread use of ORR electrocatalysts in practical applications. Model electrocatalysts may provide additional evidence to establish effective materials. Developing carbon materials doped with nitrogen heteroatoms, which have numerous carbon defects, needs to be addressed. The model electrocatalysts with precise multi-doping produce extremely active materials for ORRs in an acidic medium.

**Single-atom carbon electrocatalysts** are broadly investigated and studied materials. A systematic assessment is highly required for ORRs in an acidic and basic environment. The synthesis method for diverse single-atom electrocatalysts is very difficult, so it is necessary to develop a standard fabrication method with improved techniques to prepare the wide range of SAECs for industrial utilization. Because of their superior characteristics and inexpensive nature, the advancement of SAECs through the non-noble metals is an exciting initiative, so further study is required for synthesis of non-precious SAECs, which provides an incredible direction for future electrocatalysis. To explore the non-precious SAECs as well as their active site mechanisms, advanced methodology and theoretical calculation are required. The unique structure of SAECs is investigated through a theoretical method, which is crucial for designing the new functional electrocatalysts. The SAECs with enhanced activity, excellent stability, and inexpensive natures have inspired extensive research. Investigating SAECs with improved characteristics is crucial for electrocatalysis especially related to energy storage applications

**Metal–organic framework (MOF)-based carbon electrocatalysts** could really provide stability to electrocatalysts. The enhanced electrocatalytic performance of MOF-based nanomaterials is highly required, particularly in acid medium. Electrocatalysts prepared from the MOFs are highly effective, and their stability is essential for energy storage and conversion application. In the future, it is important to design new SAECs for ORRs. In order to commercialize the MOF electrocatalysts, these novel materials should eliminate the above difficulty and also satisfy the following rules: (a) an inexpensive raw material and preparation technique is vital; (b) remarkable ORR activity, stability and also an upscale synthesis method for the electrocatalyst are highly essential. Consequently, significant effort will be needed in the upcoming studies.

The intrinsic ORR activity and reaction mechanisms of electrocatalysts can be effectively explored and tested using the half-cell methods. These primary factors are greatly influencing the PEMFC’s efficiency, which is not studied in detail by RDE measurements nor in thin films. For the advancement of fuel cell technology with improved performance, the design and production process of electrode electrocatalysts is of utmost importance. Complex designs can increase the efficiency of active sites by encouraging the mass transfer property. Additionally, improving the reaction mechanisms can lead to enhancing the electrocatalyst activity and strengthening the performance of PEMFC technology. Overall, the development of Pt-free electrocatalysts by substituting the atomically distributed metal electrocatalysts within carbon supports is highly significant for fuel cell research and technology.

## Data Availability

Data derived from public domain resources.

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
