# Peer review of "Recent Advances and Synergistic Effects of Non-Precious Carbon-Based Nanomaterials as ORR Electrocatalysts: A Review"

_molecules, 2023, doi:10.3390/molecules28237751_

Round 1
Reviewer 1 Report
Comments and Suggestions for Authors
Understanding the mechanism of the electrocatalytic oxygen reduction reaction (ORR) is crucial for the development of new ORR electrocatalysts. In this work, L. Payattikul et al. summarize the fundamentals and mechanism of the ORR reaction and discuss the most recent strategies for improving the ORR electrocatalytic activity, including heteroatom doping, single metal atoms (M-N-C), and MOF-based carbon electrocatalysts. I strongly recommend the publication of this manuscript in Molecules. However, I would like to offer some minor suggestions for improvement:
1. In Equation 9, found on line 200 of page 5, ‘D’ should be replaced with ‘D0’. Please double-check it for accuracy.
2. In Section 3, it appears that the elementary steps of *O2 + H+ + e- → *OOH and *O + H+ + e- → *OH are missing.
3. In Equation 7, please make the necessary correction of ‘O2+’.
4. Please provide a detailed introduction of the indirect 4e- pathway in Section 3, and include this schematic pathway in Figure 1.
5. Based on the discussion in Section 5.1 and 5.2, it appears that the subtitle ‘Active Sites Engineering’ is more appropriate when compared to ‘Active Sites Mechanism’; Moreover, there is some overlap between Section 5.3 and Section 3. I would suggest reorganizing these two parts to clearly distinguish the ORR mechanisms.
Comments on the Quality of English LanguageWhile the manuscript is well-written, I recommend a thorough proofreading of the English language throughout the entire document to correct any typos.
Reviewer 2 Report
Comments and Suggestions for Authors
1. A general note on the graphs. Poor quality of some drawings, which makes it difficult to understand the information. For example, in Figure 14, the aspect ratios of the pictures are distorted.
2. The concept of combining graphs into one drawing is unclear (there is a synthesis scheme - there is no electrochemical experiment for it and vice versa).
2. In the chapter “5.3. Mechanism of Reaction at ORR Active Sites of Metal-Based Carbon Electrocatalyst” (line 275) when listing a large number of metals in M-N-C type materials, provide more links to relevant publications.
2. In chapter “6.1. Metal free Heteroatom Dopants Electrocatalyst" (line 317) when listing heteroatoms for doping, add links to relevant publications.
3. In Table 3, we recommend moving the “H2O2 yield” parameter into the text to simplify the table.
Round 2
Reviewer 2 Report
Comments and Suggestions for Authors
The authors took into account the reviewer's comments. Satya can be published in its current form